# The Disease-Modifying Effects of a Single Intra-Articular Corticosteroid Injection during the Freezing Phase of Frozen Shoulder in an Animal Model

**DOI:** 10.3390/ijms25179585

**Published:** 2024-09-04

**Authors:** Yongjin Ahn, Sun-Jae Lee, Yong Suk Moon, Yoon-Jin Lee, Jung Hyun Park, Yongmin Chun, Dong Rak Kwon, Sang Chul Lee

**Affiliations:** 1Department and Research Institute of Rehabilitation Medicine, Yonsei University College of Medicine, Seoul 03722, Republic of Korea; ahnmedic1@gmail.com; 2Department of Pathology, School of Medicine, Catholic University of Daegu, Daegu 42472, Republic of Korea; pathosjlee@cu.ac.kr; 3Department of Anatomy, School of Medicine, Catholic University of Daegu, Daegu 42472, Republic of Korea; ysmoon@cu.ac.kr; 4Department of Biochemistry, College of Medicine, Soonchunhyang University, Cheonan 31151, Republic of Korea; leeyj@sch.ac.kr; 5Department of Rehabilitation Medicine, Gangnam Severance Hospital, Yonsei University College of Medicine, Seoul 03722, Republic of Korea; rmpjh@yuhs.ac; 6Department of Orthopedic Surgery, Yonsei University College of Medicine, Seoul 03722, Republic of Korea; osmin120@yuhs.ac; 7Department of Rehabilitation Medicine, School of Medicine, Catholic University of Daegu, Daegu 42742, Republic of Korea

**Keywords:** frozen shoulder, intra-articular injection, corticosteroid, disease-modifying treatment, range of motion

## Abstract

Although frequently prescribed for frozen shoulder, it is not known if corticosteroid injections improve the course of frozen shoulder. This study aimed to assess the disease-modifying effects of an intra-articular corticosteroid administration at the freezing phase of frozen shoulder. Twenty-four Sprague-Dawley rats were divided into four groups. Their unilateral shoulders were immobilized for the first 3 days in all groups, followed by an intra-articular corticosteroid injection in Group A, an injection and the cessation of immobilization in Group B, no further intervention in Group C, and the cessation of immobilization in Group D. All rats were sacrificed in Week 3 of study, at which point the passive shoulder abduction angles were measured and the axillary recess tissues were retrieved for histological and Western blot analyses. The passive shoulder abduction angles at the time of sacrifice were 138° ± 8° (Group A), 146° ± 5° (Group B), 95° ± 11° (Group C), 132° ± 8° (Group D), and 158° ± 2° (Control). The histological assessments and Western blots showed greater fibrosis and inflammation in the groups that did not receive the corticosteroid injection (Groups C and D) compared to the corticosteroid-injected groups (Groups A and B). These findings demonstrate the anti-inflammatory and disease-modifying effects of corticosteroid injections during the freezing phase of frozen shoulder in an animal model.

## 1. Introduction

Frozen shoulder is a burdensome disease that interferes with patients’ activities of daily living by inflicting pain during joint mobilization and causing limitations in both passive and active range of motion (ROM) [1]. Three clinical phases have been documented: freezing, frozen, and thawing [2,3]. During the freezing phase, inflammatory responses occur, though full-blown fibrosis is usually not observed [4,5], which is consistent with the clinical manifestation of diffuse shoulder pain without severe limitations in ROM [6]. Arthroscopic findings also show fibrinous synovial inflammation without capsular contracture or adhesion [7]. In contrast, during the frozen phase, inflammation subsides while fibrotic changes become more salient [4,5], corresponding to progressive limitations in shoulder movement [6]. Intra-articular corticosteroids are usually prescribed during the freezing phase [8] considering their anti-inflammatory properties [9]. The clinical effect usually manifests as a transient pain reduction after injection [10,11], lasting for up to 12 weeks [12]. However, it is unknown if intra-articular corticosteroids can modify the course of disease to prevent progression to fibrosis when administered during the freezing phase.

A variety of animal models for frozen shoulder have been documented [13,14,15,16,17,18,19]. Immobilization of the unilateral shoulder joint has been shown to induce secondary frozen shoulder in these models. In the case of Sprague-Dawley rats, 3 days of immobilization resulted in inflammatory cell infiltration upon histological examination, akin to the freezing phase in human primary frozen shoulder, while 3 weeks of immobilization resulted in histological findings analogous to the frozen phase [20]. There are only a limited number of studies that performed intra-articular injections in this rat model [21,22,23], and in all of these studies, the injections were performed at or after 3 weeks of shoulder immobilization. Therefore, the consequences of an intra-articular corticosteroid injection during the freezing phase, when inflammation is predominant and fibrosis has yet to take place, have not been studied in these rat models.

This study aimed to assess the disease-modifying effects of an intra-articular corticosteroid injection, during the freezing phase, at preventing disease progression to fulminant fibrosis. We hypothesized that the intra-articular administration of corticosteroids, combined with the release of immobilization, at the freezing phase would result in the normalization of the shoulder ROM and disease pathology, including fibrosis.

## 2. Results

### 2.1. Study Design

We first determined the duration of shoulder immobilization that best simulated the freezing phase in a rat model. Although a previous report showed prominent inflammatory cell infiltration and capillary proliferation 3 days after the immobilization of the rat shoulders, which is consistent with the “freezing phase” [20], the lack of other related references and our need to precisely simulate the freezing phase prompted us to include this step. Next, we performed an intra-articular corticosteroid injection during the freezing phase to test its disease-modifying effects (Figure 1). Male 7-week old Sprague-Dawley rats were used, and the rats were allocated to the study groups in a random manner by assigning a random number to each rat. All authors were blinded to the group assignment except for the laboratory technician.

To evaluate the optimal duration of immobilization required to simulate the freezing phase, 18 rats were allocated into three groups (*n* = 6 in each), with 3, 4, or 5 days of shoulder immobilization. The unilateral shoulders were fixated in adduction and internal rotation using a molding plaster [20,22,24,25]. Following immobilization, the plasters were removed, and the rats were sacrificed by cervical dislocation. The shoulder tissues were retrieved for histological and Western blot analyses. We selected the group showing the highest expression of inflammatory markers and the lowest expression of fibrosis markers as the group most adequately simulating the freezing phase.

Next, to determine the effects of an intra-articular corticosteroid injection during the freezing phase, 24 rats were allocated into four groups (*n* = 6 in each), all of which underwent unilateral shoulder immobilization for the duration that was found to best simulate the freezing phase in the first stage of the experiment. After this period of immobilization, Group A received an intra-articular corticosteroid with continued immobilization, Group B received an intra-articular corticosteroid with a cessation of immobilization, Group C received continued immobilization without an injection, and Group D received a cessation of immobilization without an injection. Six control shoulders were retrieved from the non-immobilized shoulders. All rats were sacrificed at the third week of the study, and the tissues were prepared in the same manner as during the first stage of the experiment. Intraperitoneal anesthesia, with 40 mg/kg of tiletamine hydrochloride and zolazepam hydrochloride (Zoletil; Virbac, Carros, France) and 1.0–5.0 mg/kg of xylazine (Rompun; Bayer AG, Leverkusen, Germany), was performed during both plaster fixation and removal [22]. The intra-articular triamcinolone injection was performed from the dorsal side of the rat shoulder after palpating for the shoulder joint. A bolus of 38 μL of 20 mg/mL triamcinolone acetonide, dissolved in 0.9% normal saline, was injected with a Hamilton syringe with a 28-gauge needle for Groups A and B [22,26].

In the case of any adverse events, we had planned to document the incidence and exclude the animal from study. However, no complications or deaths occurred. Aside from the unilateral shoulder immobilization, the rats were allowed free ambulation, and food and water were provided as needed. Each rat was kept in a separate cage in the same room at 24 °C and 45% relative humidity.

### 2.2. Optimal Duration of Immobilization for Simulation of Freezing Phase

#### 2.2.1. Histological Findings

A one-way ANOVA of the capsular thickness showed no significant differences among the three groups (0.15 ± 0.06 mm for Day 3, 0.17 ± 0.04 mm for Day 4, and 0.25 ± 0.05 mm for Day 5 of immobilization, *p* = 0.163). However, the capsular thickness tended to increase with the increasing duration of immobilization. This pattern was also observed in the immunohistochemical staining for Type III Collagen, as the semi-quantitative scores tended to rise with the increasing duration of immobilization (Figure 2J–L and Figure 3D).

Conversely, the expression levels of inflammatory markers (i.e., CD68, IL-6, and TNF-α) were highest on Day 3 of immobilization and tended to decrease afterwards. (Figure 2A–I and Figure 3A–C A one-way ANOVA revealed significant differences in the semi-quantitative scores of both IL-6 and TNF-α among the three groups (*p* < 0.001 for both). Tukey’s post-hoc test showed significant differences in IL-6 expression between Days 3 and 4 (*p* < 0.001), and between Days 3 and 5 of immobilization (*p* = 0.002), while TNF-α expression was significantly different between Days 3 and 4 (*p* < 0.001), and between Days 4 and 5 of immobilization (*p* = 0.016). Although CD68 expression levels were not significantly different among the groups (*p* = 0.064), the semi-quantitative scores were highest on Day 3 and lowest on Day 5 of immobilization.

#### 2.2.2. Western Blot

A one-way ANOVA and post-hoc Duncan multiple range test showed significant differences in the expression levels of four proteins among and between the groups of varying durations of immobilization. For inflammatory markers, Day 3 of immobilization showed a significantly greater expression of both IL-6 and CD68 compared to Days 4 and 5 of immobilization (Figure 4A,B). TNF-α expression was greater on Days 3 and 4 compared to Day 5 of immobilization, and there was no significant difference between Days 3 and 4 of immobilization (Figure 4C). On the contrary, the Type III Collagen expression level on Day 3 was significantly lower compared to Days 4 and 5 of immobilization (Figure 4D), showing that fibrosis progressed as immobilization continued. The Western blot results were in accordance with the immunohistochemical staining, as Day 3 of immobilization induced the greatest expression of inflammatory markers and the lowest expression of fibrosis markers. From the above analyses, we determined that 3 days of immobilization most adequately simulated the freezing phase.

### 2.3. Efficacy of Intra-Articular Corticosteroid Injections during the Freezing Phase

#### 2.3.1. Passive Shoulder ROM

The passive shoulder abduction angle values for each group at the baseline, Day 3, and Week 3 of study, and the differences in the values before and after the intervention (Δ = (shoulder abduction angle at Week 3) − (shoulder abduction angle at Day 3)) are shown in Table 1. While Group D and the Control group did not show significant changes in the shoulder abduction angle between Day 3 and Week 3 (*p* = 0.334 and 0.880, respectively), Group A and B, which both received an intra-articular injection at Day 3, showed significant improvement in their shoulder abduction angle (*p* = 0.048 and <0.001, respectively). Group C, which continued to be immobilized, showed a significant reduction in their passive shoulder abduction angle (*p* < 0.001) (Table 1).

A one-way ANOVA of the Δ for the shoulder abduction angle values revealed significant differences among the groups (*p* < 0.001). Tukey’s post-hoc test showed significant differences between Groups A and C (*p* < 0.001), Groups B and C (*p* < 0.001), and Groups C and D (*p* < 0.001); compared with the Control group, Groups B and C showed significant differences in the Δ for their shoulder abduction angles (*p* = 0.009, and <0.001, respectively).

A one-way ANOVA of the final shoulder abduction angle values (at Week 3) revealed significant differences among the groups (*p* < 0.001). Tukey’s post-hoc test showed significant differences between Groups A and C (*p* < 0.001), Groups B and C (*p* < 0.001), Groups B and D (*p* = 0.035), and Groups C and D (*p* < 0.001); compared with the Control Group, Groups A, C, and D showed significant differences in their shoulder abduction angles at Week 3 (*p* = 0.001, <0.001, and <0.001, respectively).

#### 2.3.2. Histological Findings

The corticosteroid-injected groups (Groups A and B) showed a comparable degree of capsular thickness to the Control; capsular thickness was markedly increased in the groups that did not receive corticosteroid treatment (Groups C and D) (Figure 5A–E). A one-way ANOVA of the capsular thickness showed significant differences among the five groups (*p* < 0.001, 0.28 ± 0.09 mm for Group A; 0.21 ± 0.03 mm for Group B; 0.39 ± 0.08 mm for Group C; 0.34 ± 0.11 mm for Group D; and 0.19 ± 0.04 mm for the Control). A post-hoc Tukey’s test revealed significant differences between the Control and Group C (*p* = 0.002), the Control and Group D (*p* = 0.031), and Groups B and C (*p* = 0.004). The capsular thickness of the corticosteroid-injected groups was not significantly different from that of the Control (*p* = 0.396, 0.998, for comparison with Groups A and B, respectively).

For the examination of fibrosis, the Masson’s trichrome stain (Figure 5F–J), Vimentin immunostaining (Figure 6F–J), and Type III Collagen stain (Figure 6K–O) showed greater staining intensity in the groups that did not receive the corticosteroid injection compared to the Control or corticosteroid-injected groups. This was supported by a semi-quantitative analysis (Figure 7A,C,D). For the Masson’s trichrome stain, a one-way ANOVA showed significant differences in the semi-quantitative scores among the groups (*p* < 0.001). The post-hoc test showed significant differences between Groups A and C, and Groups B and C (*p* = 0.011 and 0.001, respectively). For the Vimentin immunostaining, a one-way ANOVA showed significant differences in the semi-quantitative scores among the groups (*p* < 0.001). The post-hoc test showed significant differences between Groups A and C, Groups B and C, Groups A and D, Groups B and D, and Groups C and D (*p* < 0.001, <0.001, 0.033, 0.012, and 0.008, respectively). For the Type III Collagen stain, a one-way ANOVA showed significant differences among the groups (*p* = 0.005). The post-hoc test showed significant differences between Groups B and C (*p* = 0.019).

For the examination of inflammatory marker expression, the IL-6 immunostaining (Figure 6A–E) showed a similar pattern among the groups as seen with the fibrosis markers, but the differences among the groups were less prominent. A one-way ANOVA of the semi-quantitative scores of the IL-6 immunostaining (Figure 7B) showed significant differences among the groups (*p* = 0.017). The post-hoc test showed a significant difference only between the Control and Group C (*p* = 0.019).

#### 2.3.3. Western Blot

Both one-way ANOVA and post-hoc Duncan multiple range testing showed significant differences in the expression levels of six proteins among and between groups (Figure 8). For the inflammatory markers (Figure 8A–D), Group B showed significantly lower levels of all four markers (i.e., IL-1α, IL-1β, TNF-α, and TNF-β) compared to Groups A, C, and D. In the case of IL-1α and IL-1β expression, Group B showed no significant difference from the Control. Group A showed significantly lower levels of all four markers compared to Group C. Also, Group A showed significantly lower levels of TNF-α and TNF-β, but showed higher levels of IL-1β compared to Group D. Finally, Group D showed significantly lower levels of IL-1α, IL-1β, and TNF-β levels compared to Group C.

For Alarmin molecules (Figure 8E,F), Group B showed significantly lower levels of the two markers (HMGB1 and RAGE) compared to Groups A, C, and D. However, the expression levels of both HMGB1 and RAGE were greater compared to the Control group. Group A showed a significantly lower level of RAGE, but not HMGB1, expression compared to Groups C and D. Finally, Group D showed a significantly lower level of RAGE, but not HMGB1, expression compared to Group C.

## 3. Discussion

The passive shoulder abduction angles were significantly improved in the corticosteroid-injected groups (Groups A and B) at Week 3 of this study compared to Day 3 of immobilization. The release of shoulder immobilization alone (Group D) did not improve the ROM, while the continuation of shoulder immobilization (Group C) resulted in the aggravation of the ROM, resulting in progression to the frozen phase. In a previous study with rat models, intra-articular corticosteroid injections, combined with cast release at Week 3 of immobilization, resulted in the normalization of the shoulder abduction angle (139.0 ± 9.6) at 2-weeks post-intervention, to a level comparable to healthy shoulders (153.0 ± 2.7) [22]. Although this result is largely in accordance with that of Group B in our study, the study implemented a longer duration of immobilization (3 weeks instead of 3 days in our study) and may suggest that an intra-articular corticosteroid combined with shoulder re-mobilization can reverse the loss of the ROM even when administered after the freezing phase at the early frozen phase. However, considering that the final shoulder abduction angle of Group B was 145.7 ± 5.2, which is closer to that of the Control, it would be more effective to deliver the injection earlier during the freezing phase.

The anti-fibrotic effects of the intra-articular corticosteroid were clearly observed from our histological assessments. The staining intensities of the Masson’s trichrome stain, Vimentin, and Type III Collagen immunohistochemistry were significantly lower in the corticosteroid-injected groups (Groups A and B) compared to those that did not receive the injection (Groups C and D), signifying the disease-modifying effects of intra-articular corticosteroids for preventing the progression of fibrosis. The mechanism of the anti-fibrotic effects of corticosteroids is not clearly elucidated, especially in musculoskeletal systems. The anti-fibrotic effects of systemic corticosteroid therapy, widely used in fibrotic lung diseases, are mainly achieved indirectly, by attenuation of inflammatory process (e.g., alveolitis) [27,28]. However, there are in vitro studies documenting the direct inhibitory effects of corticosteroids on both fibroblast proliferation and production of connective tissue matrix components [29,30]. In our study, the immunostaining of Vimentin, a fibroblast marker, showed a significantly lower expression in the corticosteroid-injected groups (Groups A and B) compared to the groups that did not receive corticosteroid injections (Groups C and D). This could be due to the direct inhibitory effects of corticosteroids on fibroblast proliferation or may be indirectly mediated by inflammatory factors. In the context of frozen shoulder, IL-17 was found to be produced by a subpopulation of T-cells in the diseased shoulders and was shown to affect fibroblast cell viability by increasing anti-apoptotic gene expression (e.g., BCL2) [31]. Therefore, intra-articular corticosteroids may inhibit fibroblast proliferation by inhibiting T-lymphocytes and the subsequent release of IL-17. Although our study did not assess IL-17 expression, we showed, by Western blot, the decreased expression of a number of inflammatory cytokines in the corticosteroid-injected groups compared to those that did not receive the corticosteroid injection. There were no significant differences in IL-1α and IL-1β expression between Group B and the Control, while Group A showed a significantly lower expression of IL-1α, IL-1β, TNF-α, TNF-β, and RAGE compared to Group C. Such a reduction in inflammatory factors could have contributed to the secondary modulation of fibroblast proliferation and activity.

The anti-fibrotic effects of the corticosteroid were also quantitatively assessed by capsular thickness measurements. In our study, capsular thickness in the corticosteroid-injected groups (Groups A and B) did not show a significant difference from that of the Control, while the groups that did not receive the injection showed a significant difference. This observation aligns with the semi-quantitative scores of the Masson’s trichrome stain, Vimentin, and Type III Collagen immunostaining (Figure 7A,C,D). In one study, which measured capsular thickness in rat models, no significant differences were observed between a group that underwent shoulder immobilization for 8 weeks and a group that received a corticosteroid injection after the 8-week period of immobilization [21], which is contrary to our findings. However, this study did not report the unit of measurement (which was millimeters in our study), preventing a direct comparison with our results. Also, the difference may have resulted from the longer duration of immobilization (8 weeks versus 3 days in our study). This also implies that an intra-articular corticosteroid may not be able to alter the course of fibrosis when injected after the frozen phase. Another study compared the effects of sham versus corticosteroid injections in rat models that were immobilized for 3 weeks and reported a greater number of synovial membrane folds and thicker membranes due to fibrosis upon a gross histological examination [22]. Thus, it is possible that an intra-articular corticosteroid injection can reduce fibrosis even when injected at the early frozen phase (e.g., week 3 of immobilization). However, this study did not include a quantitative capsular thickness measurement, making a direct comparison with our study difficult.

Clinically, an intra-articular corticosteroid is frequently prescribed in conjunction with physiotherapy. Physiotherapy alone is associated with improved external rotation and abduction ROM compared to no treatment or a placebo [32,33]. Intra-articular corticosteroids combined with physiotherapy showed a similar or greater effect compared to intra-articular corticosteroids alone, depending on references [34,35,36,37]. One study reported a lower Shoulder Pain and Disability Index score, less night pain, and greater abduction and external rotation ROM at 6 weeks after the intervention in the combined intra-articular corticosteroid and physiotherapy group versus the intra-articular corticosteroid-only group [34]. In our study, we compared Group A, analogous to the intra-articular corticosteroid-only treatment, and Group B, analogous to the combined intra-articular corticosteroid and physiotherapy treatment, as shoulder re-mobilization effectively simulates physiotherapy. In our study, no significant differences in the Δ of the shoulder abduction angle and the final shoulder abduction angle at Week 3 were found between Groups A and B. However, the final shoulder abduction angle of Group B was comparable to that of the Control, while Group A showed a significant difference from the Control. Although there were no significant differences in the capsular thickness or semi-quantitative histological scores of fibrosis markers (i.e., Masson’s trichrome stain, Vimentin, and Type III Collagen) between Groups A and B, our Western blot showed a greater expression of both inflammatory and alarmin molecules (i.e., IL-1α, IL-1β, TNF-α, TNF-β, HMGB1, and RAGE) in Group A compared to Group B. Our results align with the previous clinical report, manifesting the greater effects of the combined treatment compared to the intra-articular corticosteroid alone.

There are several limitations to our study. Firstly, as the sample size calculation was powered by the passive shoulder ROM, our study was not powered to detect differences in secondary outcomes, such as differences in histological and Western blot results. Secondly, we included only a limited number of proteins for our immunohistochemical analysis. Markers for myofibroblasts and myoendothelium, such as α-SMA and CD31, could be studied in future experiments. Also, further molecular studies are needed to elucidate the effects of the various cytokines examined in our study on fibroblasts or fibrosis in general. Finally, rat frozen shoulder models that are induced secondary to immobilization are not immaculate substitutes for the study of primary frozen shoulder in humans.

## 4. Materials and Methods

### 4.1. Measurement of Passive Shoulder ROM

Passive shoulder abduction angle was assessed as the primary outcome measure. The shoulder abduction angle was measured at baseline, during the “freezing phase”, and during the third week of study or immediately prior to sacrifice. The measurement was performed under intraperitoneal anesthesia, which is identical to the procedure during plaster fixation and removal. A total of 10 g of weight was applied to the distal end of humeral shaft, equivalent to approximately 3.92 × 10^−3^ Nm torque [20,25]. The angle between the scapular spine and humeral shaft was assessed using a goniometer. The average of three separate measurements was used for analysis [25].

### 4.2. Histological Evaluation

The shoulder tissues from all 6 rats from each group were retrieved for histological assessment. The tissues were formalin-fixed, decalcified with 10% formic acid, and paraffin-embedded. The paraffin-embedded specimens were sectioned into 5-μm-thick slices with a microtome. The prepared sections were stained in Hematoxylin and eosin (H&E) and Masson’s trichrome stain, which enables selective staining of collagen and fibrin, allowing for a more detailed view of shoulder capsule, and examined under a light microscope.

Immunohistochemical staining was performed for detection of inflammatory markers (i.e., Cluster of Differentiation 68 (CD68), Interleukin-6 (IL-6), and Tumor Necrosis Factor-α (TNF-α) and fibrosis (i.e., Type III Collagen and Vimentin). The paraffin-embedded and microtome-sectioned specimens were treated with phosphate-buffered saline (PBS) and incubated in citrate buffer for 30 min at 95 °C for antigen retrieval. Endogenous peroxidases were blocked with 0.3% hydrogen peroxide in PBS, and non-specific antibody binding was blocked with PBS with 10% horse, goat, or rabbit serum (Vector Laboratories, Newark, CA, USA) for 30 min. Sections were incubated at room temperature with the following primary antibodies at 1:100 to 1:200 dilution: Rabbit anti-CD68 polyclonal antibody (ab125212; Abcam, Cambridge, UK), mouse anti-IL-6 monoclonal antibody (ab9324; Abcam), mouse anti-TNF-α monoclonal antibody (sc-52746; Santa Cruz Biotechnology, Dallas, TX, USA), rabbit anti-Collagen III polyclonal antibody (ab7778; Abcam), and mouse anti-Vimentin monoclonal antibody (sc-6260; Santa Cruz Biotechnology). After incubation with primary antibodies, the sections were incubated with biotinylated anti-mouse or anti-rabbit IgG secondary antibodies at 1:100 dilution. The sections were then washed with PBS and treated with avidin–biotin–peroxidase complex (Vector Laboratories) and underwent peroxidase reaction with 0.05M Tris-HCl (pH 7.6). The sections were counterstained with hematoxylin.

For examination of the stained slides, Axiophot Photomicroscope (Carl Zeiss, Jena, Germany) was used. The images were stored with AxioCam MRc5 (Carl Zeiss) and digitally examined using SlideViewer (Version 2.7, 3DHISTECH, Budapest, Hungary), including the measurement of capsular thickness at axillary recess (Figure 9A,B). For each section, four fields were selected randomly and photographed for semi-quantitative analysis. Semi-quantitative scoring was done twice on separate occasions, each by two independent physicians blinded to the group allocation, one with more than 10 years of experience as a pathologist (Rater 1) and a trained physician (Rater 2). The intensity and extent of staining in Masson’s trichrome stain and immunohistochemistry were scored as 0 (negative staining), 1 (weak but detectable), 2 (mildly positive), 3 (moderately positive), or 4 (strongly positive) [22,38,39].

### 4.3. Western Blot

Western blot was performed for detection of inflammatory proteins (i.e., CD68, IL-1α, IL-1β, IL-6, TNF-α, and TNF-β), fibrosis markers (i.e., Type III Collagen), and Alarmin molecules (high mobility group box 1 (HMGB1), and receptor for advanced glycation end products (RAGE)). Approximately 2–3 mm^3^ of shoulder tissues were retrieved from 3 out of 6 rats in each group in both the first and second stage of experiments for Western blot. The tissue samples were homogenized and denatured using Laemmli buffer. Proteins were separated by order of molecular weight by Sodium dodecyl sulfate-polyacrylamide gel electrophoresis using NUPAGE MOPS SDS running buffer (NP0001; Thermo Fisher Scientific, Waltham, MA, USA). The proteins were transferred to PVDF membrane (10600023; Amersham Cytiva, Amersham, UK) and blocked with casein blocking buffer in PBS (37528; Thermo Fisher Scientific). The PVDF membrane was then incubated with primary antibodies: Rabbit anti-CD68 polyclonal antibody (1:500, ab125212; Abcam, Cambridge, UK), mouse anti-IL-1α monoclonal antibody (1:500, ab239517; Abcam), rabbit anti-IL-1β polyclonal antibody (1:500, ab1832P; Merck Millipore, Burlington, MA, USA), mouse anti-IL-6 monoclonal antibody (1:500, ab9324; Abcam), mouse anti-TNF-α monoclonal antibody (1:500, sc-52746; Santa Cruz Biotechnology), rabbit anti-TNF-β polyclonal antibody (1:500, PA5-116055; Thermo Fisher Scientific), rabbit anti-Collagen III polyclonal antibody (COL III) (1:500, ab7778; Abcam). Rabbit anti-HMGB1 monoclonal antibody (1:500, ab79823; Abcam), and anti-RAGE polyclonal antibody (1:500, ab37647; Abcam). After incubation with primary antibodies, the PVDF membrane was incubated with anti-rabbit (1:500, LF SA8002; AbFrontier, Seoul, Republic of Korea) and anti-mouse (1:500, 7076S; Cell Signaling, Danvers, MA, USA) secondary antibodies conjugated with horseradish peroxidase. Chemiluminescence reagents were used to visualize the secondary antibodies. For quantification of relative protein band densities, TINA software (version 2.10e, Raytest Isotopenmessgeraete, Straubenhardt, Germany) was used.

### 4.4. Statistical Analysis

Statistical analysis was performed using R version 4.3.1 software (R Foundation, Vienna, Austria). Shapiro–Wilk test was used to demonstrate normal distribution of continuous outcome variables (i.e., passive shoulder abduction angle). Paired *t*-test was used for within-group comparisons of passive shoulder abduction angles at Day 3 and Week 3 of the second experiment. One-way analysis of variance (ANOVA) was used for among-group comparisons of quantitative variables, after which Tukey’s post-hoc test or Duncan multiple range test was used for between-group comparisons in the case of one-way ANOVA results that showed statistical significance. Statistical significance was determined as *p* < 0.05. Weighted Kappa was used to assess the intra- and inter-observer reliability of semi-quantitative scoring: intra-rater reliability was calculated as 0.881, showing strong agreement, while inter-rater reliability was 0.793, showing substantial agreement.

## 5. Conclusions

This study demonstrated the disease-modifying effects of an intra-articular corticosteroid delivered during the freezing phase of frozen shoulder. The intra-articular corticosteroid injection significantly improved the passive ROM while preventing both progression of fibrosis and fibroblast proliferation, and these effects were potentiated by the release of immobilization, allowing for free shoulder movement. The anti-fibrotic and ROM-enhancing properties should be considered when prescribing intra-articular corticosteroids, which may not only be symptom-relieving, but also disease-modifying when administered during the freezing phase. Further clinical studies should follow in order to verify the disease-modifying effects of corticosteroids in human subjects.

## Figures and Tables

**Figure 1 ijms-25-09585-f001:**
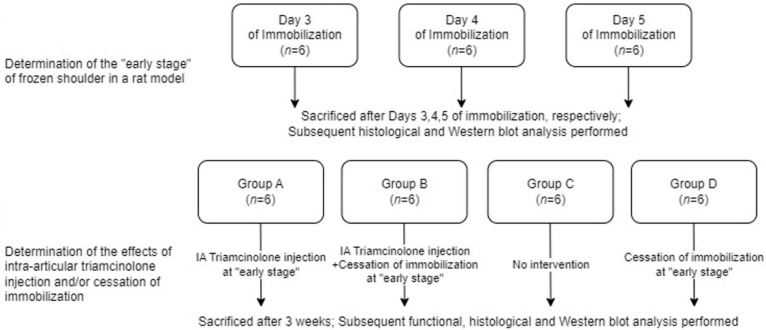
Illustration of the study protocol.

**Figure 2 ijms-25-09585-f002:**
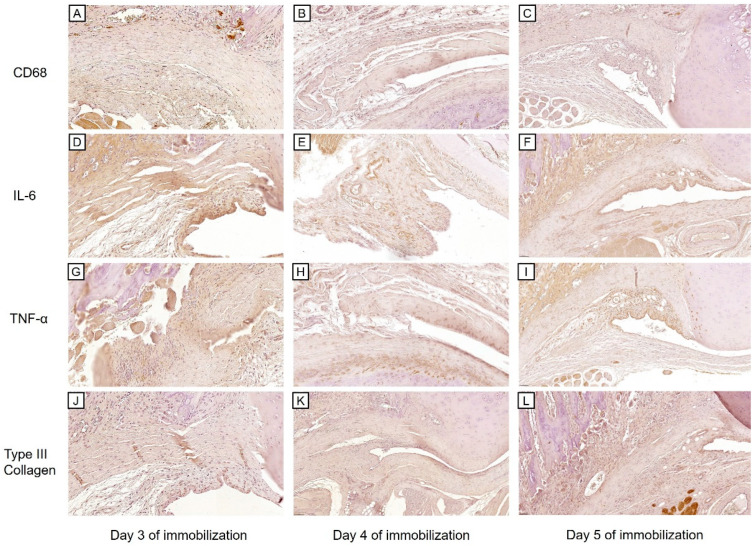
Immunohistochemical findings on Days 3–5 of immobilization. Images show the synovium and subsynovial structures of axillary recess. (**A**–**C**) CD68, (**D**–**F**) IL-6, (**G**–**I**) TNF-α, (**J**–**L**) Type III Collagen immunohistochemical staining on Days 3–5 of immobilization (×40).

**Figure 3 ijms-25-09585-f003:**
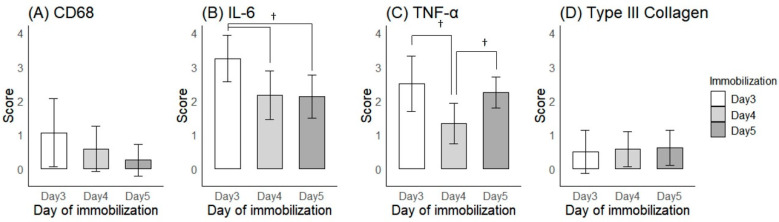
Semi-quantitative scores of immunohistochemical findings. (**A**) CD68, (**B**) IL-6, (**C**) TNF-α and (**D**) Type III Collagen on Days 3–5 of immobilization. † indicates *p* < 0.05 upon post-hoc testing between groups.

**Figure 4 ijms-25-09585-f004:**
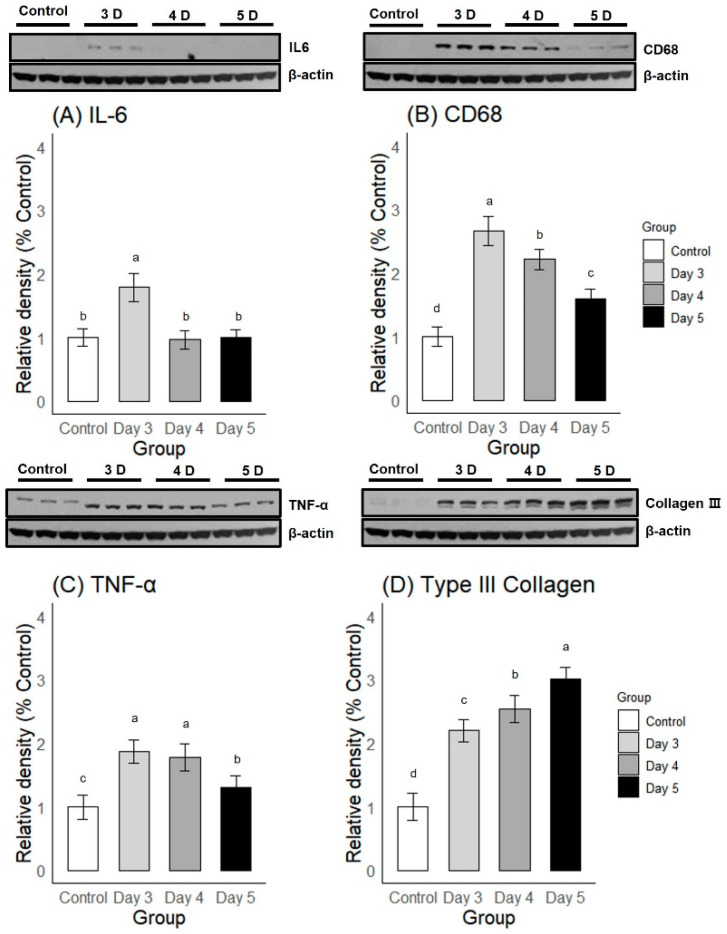
Western blot results of shoulder tissues with varying durations of immobilization. Relative densities of protein bands were compared with one-way ANOVA and post-hoc Duncan multiple range test. Different letters (a–d) on the bar represent significant differences upon Duncan multiple range testing.

**Figure 5 ijms-25-09585-f005:**
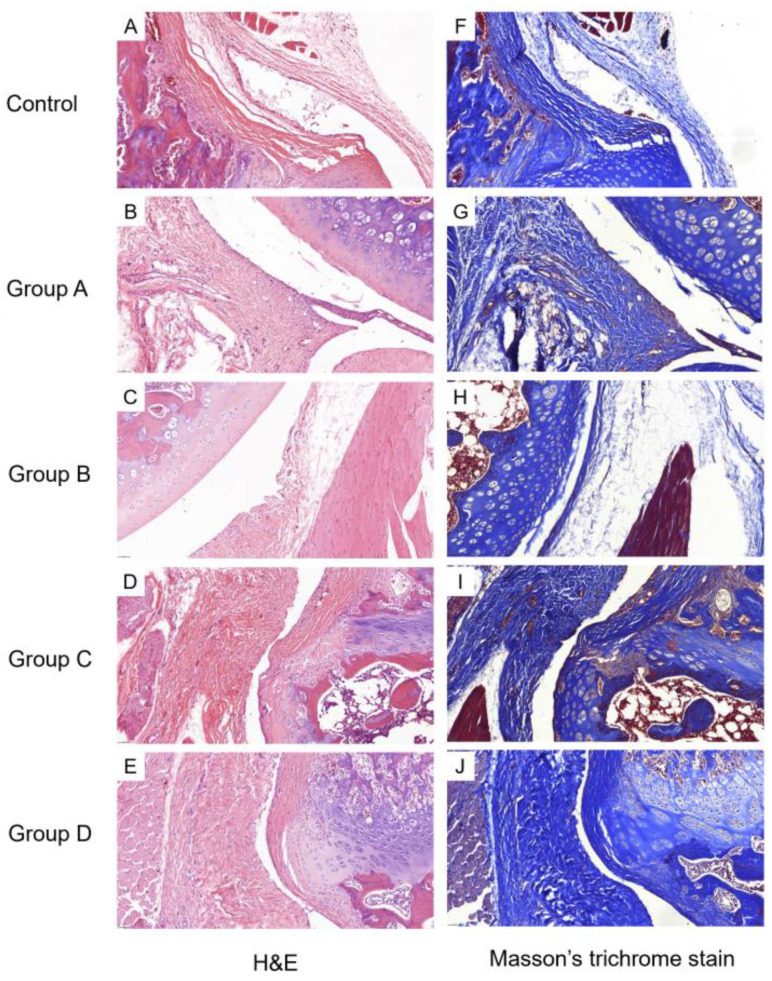
Staining of Control and Groups A–D. Images show the synovium and subsynovial structures of axillary recess. (**A**–**E**) H&E and (**F**–**J**) Masson’s trichrome stain of Control and Groups A–D (×40). Group A—intra-articular triamcinolone only; Group B—intra-articular triamcinolone + cessation of immobilization; Group C—no intervention; Group D—cessation of immobilization only.

**Figure 6 ijms-25-09585-f006:**
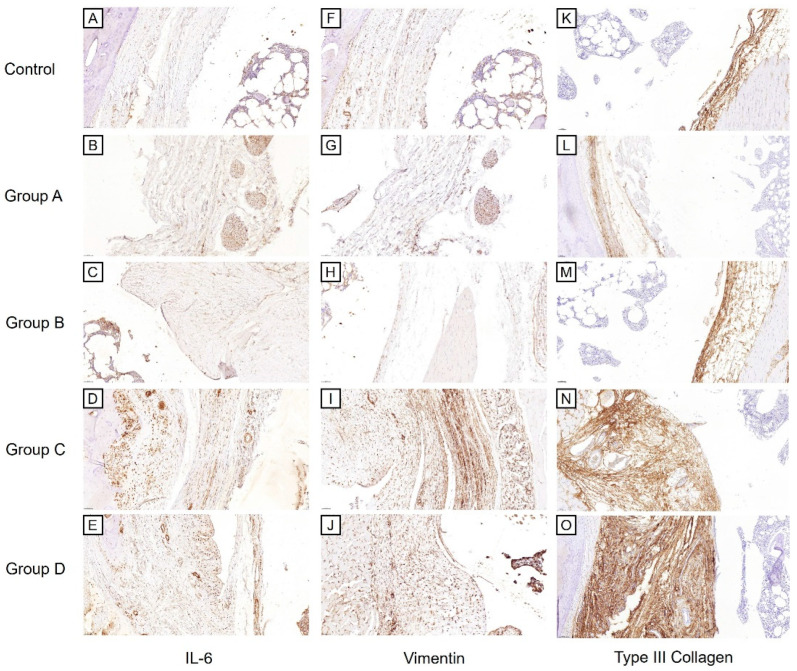
Immunohistochemical findings of Control and Groups A–D. The images show the synovium and subsynovial structures of axillary recess. (**A**–**E**) IL-6, (**F**–**J**) Vimentin, and (**K**–**O**) Type III Collagen immunohistochemical staining of Control and Groups A–D (×40). Group A—intra-articular triamcinolone only; Group B—intra-articular triamcinolone + cessation of immobilization; Group C—no intervention; Group D—Cessation of immobilization only.

**Figure 7 ijms-25-09585-f007:**
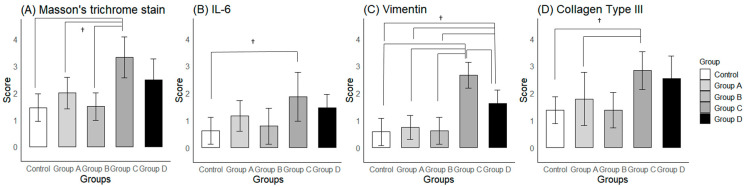
Semi-quantitative scores of histological and immunohistochemical findings. (**A**) Masson’s trichrome stain, (**B**) IL-6 immunostaining, (**C**) Vimentin, and (**D**) Type III Collagen immunostaining for Control and Groups A–D. Group A—intra-articular triamcinolone only; Group B—intra-articular triamcinolone + cessation of immobilization; Group C—no intervention; Group D—cessation of immobilization only. † indicates *p* < 0.05 upon post-hoc testing between groups.

**Figure 8 ijms-25-09585-f008:**
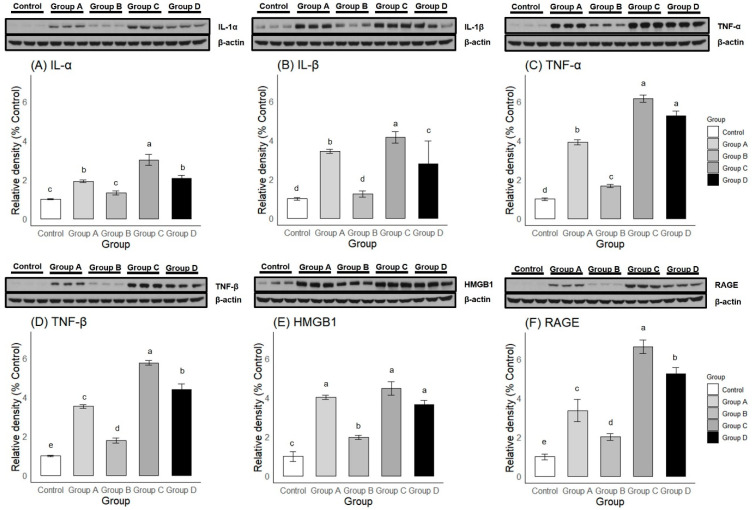
Western blot results of (**A**) IL-1α, (**B**) IL-1β, (**C**) TNF-α, (**D**) TNF-β, (**E**) HMGB1, and (**F**) RAGE post-intervention during the freezing phase. Group A—intra-articular triamcinolone only; Group B—intra-articular triamcinolone + cessation of immobilization; Group C—no intervention; Group D—cessation of immobilization only. Relative densities of protein bands were compared with one-way ANOVA and post-hoc Duncan multiple range test. Different letters (a–e) on the bar represent significant differences upon Duncan multiple range testing.

**Figure 9 ijms-25-09585-f009:**
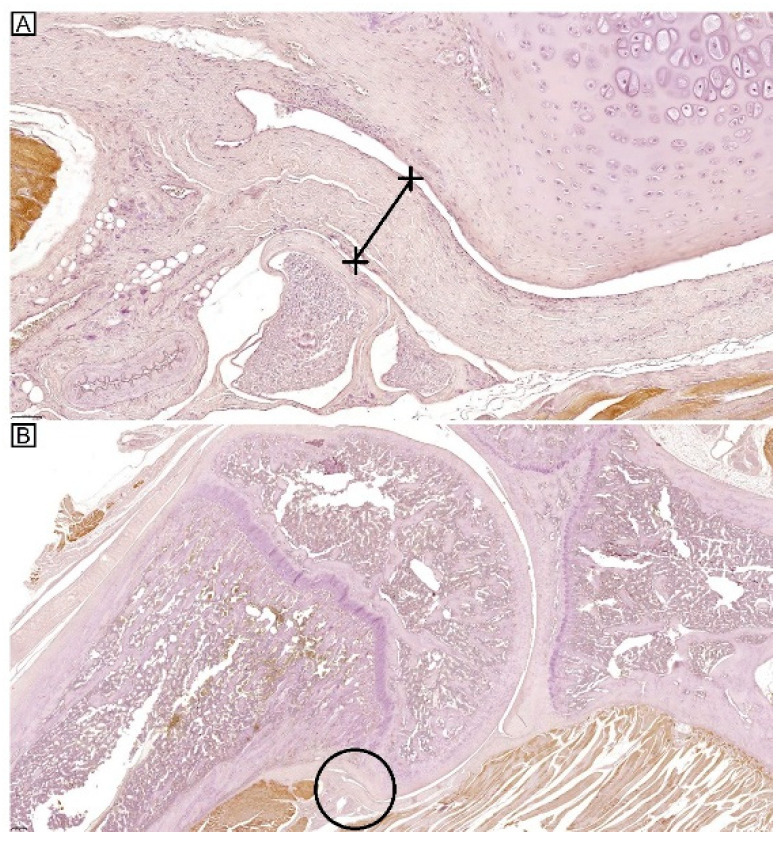
Measurement of capsular thickness at the axillary recess. (**A**) at ×40 magnification, the capsular thickness measurement is displayed as a straight line, (**B**) at ×5 magnification, the axillary recess at which the measurement was made is marked in circle.

**Table 1 ijms-25-09585-t001:** Passive shoulder abduction angle at Day 3 and Week 3 of study.

		Passive Shoulder Abduction Angle			
	Baseline	Day 3	Week 3	Δ	*P*
Group A	159 ± 3	135 ± 6	138 ± 8	3 ± 3	0.048
Group B	160 ± 3	135 ± 4	146 ± 5	11 ± 4	<0.001
Group C	159 ± 4	135 ± 5	95 ± 11	−40 ± 7	<0.001
Group D	158 ± 5	129 ± 2	132 ± 8	3 ± 7	0.334
Control	158 ± 4	158 ± 3	158 ± 2	−0 ± 3	0.880

Differences between shoulder abduction angles at Day 3 and at Week 3 are presented as Δ, and P values of the paired *t*-test performed between shoulder abduction angles at Day 3 and at Week 3 are shown. Group A—intra-articular triamcinolone only; Group B—intra-articular triamcinolone + cessation of immobilization; Group C—no intervention; Group D—cessation of immobilization only.

## Data Availability

The data presented in this study are available upon request from the corresponding author.

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
