# Peer review of "The Disease-Modifying Effects of a Single Intra-Articular Corticosteroid Injection during the Freezing Phase of Frozen Shoulder in an Animal Model"

_ijms, 2024, doi:10.3390/ijms25179585_

Round 1

Reviewer 1 Report

Comments and Suggestions for Authors

I congratulate the authors on a very carefully conducted study and an overall well written paper.  This is an excellent paper. 

Some minor suggestions:-

      1)      Try to avoid using abbreviations as much as possible, the readers find it difficult to follow the text unless those abbreviations are embedded into his/her head.   

      2)     The captions on the figures need to include details of what the various groups involved, in other words, groups A – D.  Otherwise, the reader has to go back to the text to try to work out what is going on. 

            3)   I don’t understand the term “MTS” and  I don’t understand the term “SAA”

      4)      Line 422 – The authors write “This property should be considered” please expand what you mean by “This” property.

      5)      Table 1 – “Passive SAA” I assume this is some sort of range of measurement, again, it needs to be clearly defined what is SAA. 

      6)      You report that the results for the figures are 0.1 of a SAA.  My question is, are your measurements accurate to that level?  If not, I recommend rounding out those numbers.

Author Response

Dear reviewer,

We appreciate your insightful comments. We addressed each of your comment below and revised our manuscript accordingly.

Comment 1: Try to avoid using abbreviations as much as possible, the readers find it difficult to follow the text unless those abbreviations are embedded into his/her head.

Response 1: Following the reviewer’s recommendation, we decided to avoid using abbreviations as much as possible.

Before: “MTS”, “SAA”, “IA”

After: “Masson’s trichrome stain”, “Shoulder abduction angle”, “Intra-articular”

Comment 2: The captions on the figures need to include details of what the various groups involved, in other words, groups A – D.  Otherwise, the reader has to go back to the text to try to work out what is going on.

Response 2: We added short descriptions of interventions performed for each group in the captions of relevant tables and figures.

Before:

Caption of [Table 1]

The differences between SAA at day 3 and at week 3 are presented as ΔSAA, and P values of the paired t-test performed between the SAAs at day 3 and at week 3 are shown.

Caption of [Figure 5]

Staining of Control and Groups A-D. The images show the synovium and subsynovial structures of axillary recess. (A-E) H&E, (F-J) MTS of Control and Groups A-D (×40).

Caption of [Figure 6]

Immunohistochemical findings of Control and Groups A-D. The images show the synovium and subsynovial structures of axillary recess. (A-E) IL-6, (F-J) Vimentin, (K-O) Type III Collagen immunohistochemical staining of Control and Groups A-D (×40).

Caption of [Figure 7]

Semi-quantitative scores of histological and immunohistochemical findings. (A) MTS, (B) IL-6 immunostaining, (C) Vimentin, (D) Type III Collagen immunostaining for Control and Groups A-D.

Caption of [Figure 8]

Western blot results of (A) IL-1α, (B) IL-1β, (C) TNF-α, (D) TNF-β, (E) HMGB1, and (F) RAGE post-intervention at the freezing phase. Group A – intra-articular triamcinolone only; Group B – intra-articular triamcinolone + cessation of immobilization; Group C – no intervention; Group D – Cessation of immobilization only.

After:

Caption of [Table 1]

The differences between shoulder abduction angle at day 3 and at week 3 are presented as Δ, and P values of the paired t-test performed between the shoulder abduction angles at day 3 and at week 3 are shown. Group A – intra-articular triamcinolone only; Group B – intra-articular triamcinolone + cessation of immobilization; Group C – no intervention; Group D – Cessation of immobilization only.

Caption of [Figure 5]

Staining of Control and Groups A-D. The images show the synovium and subsynovial structures of axillary recess. (A-E) H&E, (F-J) Masson’s trichrome stain of Control and Groups A-D (×40). Group A – intra-articular triamcinolone only; Group B – intra-articular triamcinolone + cessation of immobilization; Group C – no intervention; Group D – Cessation of immobilization only.

Caption of [Figure 6]

Immunohistochemical findings of Control and Groups A-D. The images show the synovium and subsynovial structures of axillary recess. (A-E) IL-6, (F-J) Vimentin, (K-O) Type III Collagen immunohistochemical staining of Control and Groups A-D (×40). Group A – intra-articular triamcinolone only; Group B – intra-articular triamcinolone + cessation of immobilization; Group C – no intervention; Group D – Cessation of immobilization only.

Caption of [Figure 7]

Semi-quantitative scores of histological and immunohistochemical findings. (A) Masson’s trichrome stain, (B) IL-6 immunostaining, (C) Vimentin, (D) Type III Collagen immunostaining for Control and Groups A-D. Group A – intra-articular triamcinolone only; Group B – intra-articular triamcinolone + cessation of immobilization; Group C – no intervention; Group D – Cessation of immobilization only.

Caption of [Figure 8]

Western blot results of (A) IL-1α, (B) IL-1β, (C) TNF-α, (D) TNF-β, (E) HMGB1, and (F) RAGE post-intervention at the freezing phase. Group A – intra-articular triamcinolone only; Group B – intra-articular triamcinolone + cessation of immobilization; Group C – no intervention; Group D – Cessation of immobilization only.

Comment 3: I don’t understand the term “MTS” and  I don’t understand the term “SAA”

Response 3: We changed “MTS” to “Masson’s trichrome stain” and “SAA” to “shoulder abduction angle” each time the term appears on the text. We changed the terms in the figures and tables as well.

Before: “MTS”, “SAA”

After: “Masson’s trichrome stain”, “Shoulder abduction angle”

Comment 4: Line 422 – The authors write “This property should be considered” please expand what you mean by “This” property.

Response 4: We changed the expression so as to be more specific with the properties of intra-articular corticosteroid injection.

Before: This property should be considered when prescribing intra-articular corticosteroid, which may not only be symptom-relieving, but also disease-modifying when administered at the freezing phase.

After: The anti-fibrotic and ROM-enhancing properties should be considered when prescribing intra-articular corticosteroid, which may not only be symptom-relieving, but also disease-modifying when administered at the freezing phase.

Comment 5: Table 1 – “Passive SAA” I assume this is some sort of range of measurement, again, it needs to be clearly defined what is SAA.

Response 5: As in “Response 3”, we opted not to use abbreviations for SAA.

Before: “SAA”

After: “Shoulder abduction angle”

Comment 6: You report that the results for the figures are 0.1 of a SAA. My question is, are your measurements accurate to that level?  If not, I recommend rounding out those numbers.

Response 6: We agree that rounding out the numbers for shoulder abduction angle (SAA) to the nearest integer is more appropriate. We changed the numbers accordingly.

Before:

[Abstract]

Passive shoulder abduction angles at sacrifice were 138.0° ± 7.8° (Group A), 145.7° ± 5.2° (Group B), 94.8° ± 11.2° (Group C), 132.2° ± 8.1° (Group D), and 157.8° ± 2.3° (Control).

[Table 1]

Table 1. Passive SAA at Day 3 and Week 3 of study.

Passive SAA

SAA at baseline

SAA at Day 3

SAA at Week 3

ΔSAA

P

Group A

159.3 ± 3.1

135.0 ± 5.8

138.0 ± 7.8

3.0 ± 2.8

0.048

Group B

160.2 ± 3.4

134.8 ± 3.5

145.7 ± 5.2

10.8 ± 3.8

<0.001

Group C

159.0 ± 4.0

135.2 ± 4.8

94.8 ± 11.2

-40.3 ± 7.2

<0.001

Group D

158.0 ± 4.8

129.0 ± 2.4

132.2 ± 8.1

3.2 ± 7.3

0.334

Control

157.5 ± 3.9

158.0 ± 3.2

157.8 ± 2.3

-0.2 ± 2.6

0.880

After:

[Abstract]

Passive shoulder abduction angles at sacrifice were 138° ± 8° (Group A), 146° ± 5° (Group B), 95° ± 11° (Group C), 132° ± 8° (Group D), and 158° ± 2° (Control).

[Table 1]

Table 1. Passive shoulder abduction angle at Day 3 and Week 3 of study.

Passive shoulder abduction angle

Baseline

Day 3

Week 3

Δ

P

Group A

159 ± 3

135 ± 6

138 ± 8

3 ± 3

0.048

Group B

160 ± 3

135 ± 4

146 ± 5

11 ± 4

<0.001

Group C

159 ± 4

135 ± 5

95 ± 11

-40 ± 7

<0.001

Group D

158 ± 5

129 ± 2

132 ± 8

3 ± 7

0.334

Control

158 ± 4

158 ± 3

158 ± 2

-0 ± 3

0.880

Reviewer 2 Report

Comments and Suggestions for Authors

ijms-3145380

Title: The disease-modifying effects of intra-articular corticosteroid injection at the freezing phase of frozen shoulder in an animal model

 In the present study the disease-modifying effects of IA corticosteroid injection 63 during the freezing phase at preventing disease progression to fulminant fibrosis were investigated in rats.

Unilateral shoulders were immobilized for 3 days in all groups, followed by intra-articular (IA) corticosteroid injection with and without cessation of immobilization or without corticosteroid injection with either no further intervention or cessation of immobilization. After 3 weeks abduction angles were measured, and axillary recess tissues were retrieved for histological and western blot analyses. Both groups which received IA injection at day 3, showed significant improvement in passive shoulder ROM. Corticosteroid-injected groups showed comparable degree of capsular thickness with control, which was markedly increased in groups which did not receive corticosteroid treatment. Also examination of fibrosis showed less staining intensity for vimentin and collagen III in corticosteroid groups compared to the other groups. The authors concluded that their findings demonstrated the long-term anti-inflammatory and disease-modifying effects of corticosteroid injection at the freezing phase of frozen shoulder.

 Strength

The question is well defined and the results provide an advance in current knowledge.

The study is correctly designed, especially with the evaluation of the study control by testing the optimal immobilization duration in order to simulate the freezing phase.

The experiment design and the methods used are clearly described. The data and analyses are presented appropriately. The results are interpreted appropriately. The article is written in an appropriate way.

Specific comments:

 Please add to the title that a single injection was used:

The disease-modifying effects of a single intra-articular corticosteroid injection at the freezing phase of frozen shoulder in an animal model’

In the abstract, you conclude that your ‘findings demonstrated the long-term anti-inflammatory and disease-modifying effects of corticosteroid injection at the freezing phase of frozen shoulder.’

Please add that this is in the animal model. Please be much more careful in your wording when translating this to the human situation.

Also: Can you speak of a long-term anti-inflammatory and disease-modifying effect after 3 weeks of follow-up? Please be much more careful with your wording.

In line 103 you define the dosage and the volume of the corticosteroid solution used. Was it dissolved in saline? Please add.

Please describe exactly where you placed the injection.

What size needle was used for the injection?

Line 191 – please describe MTS at its first use (explained in line 345)

Author Response

Dear reviewer,

We appreciate your insightful comments. We addressed each of your comment below and revised our manuscript accordingly.

Comment 1:  Please add to the title that a single injection was used:

‘The disease-modifying effects of a single intra-articular corticosteroid injection at the freezing phase of frozen shoulder in an animal model’

Response 1: We changed the title as recommended.

Before: The disease-modifying effects of intra-articular corticosteroid injection at the freezing phase of frozen shoulder in an animal model

After: The disease-modifying effects of a single intra-articular corticosteroid injection at the freezing phase of frozen shoulder in an animal model

Comment 2: In the abstract, you conclude that your ‘findings demonstrated the long-term anti-inflammatory and disease-modifying effects of corticosteroid injection at the freezing phase of frozen shoulder.’

Please add that this is in the animal model. Please be much more careful in your wording when translating this to the human situation.

 Also: Can you speak of a long-term anti-inflammatory and disease-modifying effect after 3 weeks of follow-up? Please be much more careful with your wording.

Response 2: We specified that the results are specific for an animal model. Also, we removed the expression “long-term” as we did not follow up after 3 weeks.

Before: These findings demonstrated the long-term anti-inflammatory and disease-modifying effects of corticosteroid injection at the freezing phase of frozen shoulder.

After: These findings demonstrated the anti-inflammatory and disease-modifying effects of corticosteroid injection at the freezing phase of frozen shoulder in an animal model.

Comment 3: In line 103 you define the dosage and the volume of the corticosteroid solution used. Was it dissolved in saline? Please add.

Response 3: Triamcinolone was dissolved in 0.9% normal saline, in accordance with the clinical practice in humans.

Before: 38μL of 20 mg/ml triamcinolone acetonide was injected for Groups A and B

After: Intra-articular triamcinolone injection was performed from the dorsal side of the rat shoulder after palpating for the shoulder joint. 38μL of 20 mg/ml triamcinolone acetonide dissolved in 0.9% normal saline was injected with Hamilton syringe with 28-gauge needle for Groups A and B.

Comment 4: Please describe exactly where you placed the injection.

What size needle was used for the injection?

Response 4: After palpating for the shoulder joint, intra-articular injection was performed from the dorsal side. Hamilton syringe with 28-gauge needle was used. We added the details to the manuscript.

Before: 38μL of 20 mg/ml triamcinolone acetonide was injected for Groups A and B

After: Intra-articular triamcinolone injection was performed from the dorsal side of the rat shoulder after palpating for the shoulder joint. 38μL of 20 mg/ml triamcinolone acetonide dissolved in 0.9% normal saline was injected with Hamilton syringe with 28-gauge needle for Groups A and B.

Comment 5: Line 191 – please describe MTS at its first use (explained in line 345)

Response 5: We opted not to use abbreviation for Masson’s trichrome stain (MTS) in the entire manuscript. Also, we added a brief description of Masson’s trichrome stain in the methods section.

Before: The prepared sections were stained in Hematoxylin and eosin (H&E) and Masson’s trichrome stain (MTS), and examined under a light microscope.

After: The prepared sections were stained in Hematoxylin and eosin (H&E) and Masson’s trichrome stain which enables selective staining of collagen and fibrin allowing for a more detailed view of shoulder capsule, and examined under a light microscope.